# Impact of iSupport on improving knowledge on dementia and dementia care among family caregivers to persons with dementia and formal caregivers in Bangladesh: Protocol for a non-randomized feasibility study

**Muhammed Nazmul Islam**[1][*], **Parsa Musarrat**[1], **Hanne Konradsen**[2,3], **Marie Tyrrell**[3,4], **Antara Roy**[1], **Asibul Islam Anik**[1], **Zarina Nahar Kabir**[3]

**1** Research and Evaluation Department, SAJIDA Foundation, Dhaka, Bangladesh, **2** North Zealand Hospital, Hillerod, Denmark, **3** Department of Neurobiology, Care Sciences and Society, Karolinska Institutet, Huddinge, Sweden, **4** Sophiahemmet University, Stockholm, Sweden

☯ These authors contributed equally to this work.
* nazmulislam@sajida.org

## Abstract

Dementia is a growing global public health concern, particularly in LMICs like Bangladesh. With population aging, there is an increasing prevalence of dementia; care facilities are limited. This scarcity of services places a significant responsibility on untrained family caregivers, impacting the health of persons with dementia and their caregivers. Addressing these challenges is crucial for improving dementia care, especially in resource-constrained settings. Recognizing the urgency, the WHO developed an online self-training manual, iSupport, emphasizing education, skill development, and support for caregivers. This study aims to test feasibility of iSupport i) in improving knowledge about dementia and dementia care among family and formal caregivers, and ii) reducing family caregivers' stress and improving their caregiving skills, mental health, quality of life and relationship with the person with dementia cared for. The study protocol was approved by the Institutional Review Board of SAJIDA Foundation (2024–005-SFIRB). This non-randomized feasibility study aims to recruit 44 family caregivers and 132 formal caregivers from Dhaka, Bangladesh, using a purposive sampling technique. Participants will be provided access to iSupport for two months. Pre- and post-intervention surveys will be conducted to assess changes in knowledge about dementia care among both family and formal caregivers, as well as changes in psychological well-being among family caregivers. To assess knowledge retention, a follow-up survey will be conducted. Any differences in the outcomes from the pre-intervention period will be tested against the null hypothesis of zero mean difference using a two-sample paired t-test. Additionally, multivariable regression techniques will also be used to control for possible confounders. To understand

**Data availability statement:** No datasets were generated or analyzed during the current study. Deidentified research data will be made publicly available when the study is completed and published.

**Funding:** The author(s) received no specific funding for this work.

**Competing interests:** The authors have no conflict of interest to declare.

the facilitators and barriers of implementing iSupport, a qualitative approach will be used to interview 15 family caregivers and conduct four FGDs with formal caregivers. Qualitative data will be analysed using an inductive content analysis approach.

## Introduction

Dementia is a syndrome which describes major neurocognitive disorders of the brain and is one of the main causes of disability, dependency, and death among the aging population worldwide [1,2]. Dementia affects memory, cognitive function, behaviour, and ultimately impedes a person's ability to independently carry out day-to-day tasks. According to the World Health Organization (WHO), more than 55 million people across the world have dementia and 10 million new cases are expected each year [3,4]. More than 60% of persons with dementia (PWD) are in low- and middle-income countries (LMICs) [5].

As many LMICs, Bangladesh is also experiencing population aging [6]. Life expectancy of the population of Bangladesh has increased from 71.6 years in 2016 to 72.6 years in 2019 [7]. According to recent Bangladesh Bureau of Statistics (BBS) data, as of 2022, 9.28% of the population (approximately 15.31 million people) in Bangladesh were 60 years or above [8]. Population growth and aging population has raised concerns regarding the increasing prevalence of dementia [5,9]. The exact prevalence of dementia in Bangladesh is unknown. Studies conducted in Bangladesh reported prevalence of dementia to be between 3.6% [10] and 8% [11].

In LMICs, such as Bangladesh, care and support facilities for PWD are severely lacking, especially for PWD belonging to lower socioeconomic groups [1,12,13]. For most PWD, family members with little or no knowledge of dementia and no professional training become the primary caregivers. In multiple studies, South Asian family caregivers expressed their lack of knowledge about dementia, including difficulty in recognising the symptoms of dementia [14]. Such circumstances can deprive PWD of evidence-based care and may even lead to adverse health consequences [15].

Aging is associated with declining health as the body's systems deteriorate and chronic conditions develop, underscoring the importance of adopting and maintaining healthy lifestyle practices throughout the aging process [16]. Cognitive decline in PWD may impact the person's activities of daily living, e.g., nutritional intake, eating and sleeping patterns, or continence [17]. The ability of the PWD to communicate may also be impacted by this progressive disease. The potential challenges, particularly relating to communication, experienced by the PWD necessitate their caregivers to possess knowledge and skills to manage and support the well-being of the PWD and to ensure a safe environment for them [18,19].

In recognizing dementia as a priority in public health, the WHO has developed the Global Action Plan on the Public Health Response to Dementia 2017–2025 [1]. Highlighting the increase in inequality due to the disproportionate increase in dementia in LMICs, the WHO's plan focuses on educating people on dementia, and developing skills for handling PWD through sustainable non-pharmacological interventions [1].

This plan also highlights the burden on caregivers of PWD and the need to promote their own self-care. Caregivers caring for PWD are subjected to stressful situations, making them more susceptible to developing depression and anxiety, and other mental health, and even physical health, conditions which may impact negatively on their caregiving [15,20,21].

Following the Global Action Plan, WHO has developed an online training and support manual called iSupport [1,22]. This internet-based self-help tool is designed for caregivers to improve knowledge on what dementia is, its impact and how to care for PWD and to improve caregiving skills, while also ensuring the caregiver's own well-being [22]. The iSupport manual addresses the need to ensure the well-being of the caregivers as the impact of dementia on the body, mind and social relationships aren't limited to PWD but can impact their caregivers as well [23].

The iSupport manual has been adapted in more than 20 countries around the world [4]. Since iSupport is a relatively recently developed intervention, limited evidence is available about its impact on caregivers' knowledge related to dementia, especially in LMICs. Studies conducted in Brazil and Portugal have found that the iSupport manual is both useful and meets caregivers' needs and indicated caregivers' satisfaction with the manual [24,25]. A study on the effectiveness of the iSupport manual in India reported significantly higher positive attitude of caregivers towards PWD; however, it failed to find significant differences in caregivers' perceived burden and depression, possibly owing to low adherence issues [26].

Studies on psychoeducation interventions have shown significant improvements in stress levels and quality of life of caregivers [23,27,28]. Moreover, internet-based interventions are expected to have higher efficacy than in-person interventions as they can bypass limitations in LMICs, such as lack of trained professionals and infrastructure to provide in-person training [29]. In addition, internet-based interventions are expected to help bypass stigma and other social barriers that are typically associated with seeking mental health support [25]. Furthermore, the web-based feature of the iSupport manual contributes to the overall convenience of accessing the manual for caregivers with access to the internet and compatible devices.

Most literature related to the iSupport intervention currently available are on user feedback [24,30] and process of culturally adapting the training manual [31–36]. There is a dearth of literature that evaluates interventions designed to provide caregiver support in Bangladesh. This study will be the first study evaluating the iSupport manual in Bangladesh. It will also be one of the few studies investigating the impact of the iSupport manual on formal caregivers.

## Objectives

The primary objective of this study is to assess the feasibility of iSupport manual in improving knowledge in dementia and dementia care among family and formal caregivers in Bangladesh. Additionally, this study also aims to assess whether iSupport helps family caregivers in their caregiving skills, reducing stress and improving mental health when providing care to PWD living at home as well as improving their relationship with the PWD.

## Research questions

The primary research question for family caregivers is:

1.  Does iSupport improve family caregivers' overall knowledge about dementia?

The secondary research questions for family caregivers are:

1. Does iSupport improve family caregivers' caregiving skills required to care for PWD?

2. Does iSupport reduce family caregivers' stress?

3. Does iSupport improve family caregivers' mental health?

4. Does iSupport improve family caregivers' quality of life?

5. Does iSupport improve family caregivers' relationship with PWD?

The primary research question for formal caregivers is:

1. Does iSupport improve formal caregivers' overall knowledge about dementia?

Secondary research questions for formal caregivers are:

1. Does the impact of iSupport on knowledge about dementia differ between male and female formal caregivers?

2. Does the impact of iSupport on knowledge about dementia vary with the level of caregiving skills of formal caregivers?

In addition, this study aims to capture caregivers' experience of using iSupport in relation to providing care for PWD, the perceived benefits and challenges of using iSupport, and recommendations for improving the iSupport intervention through qualitative methods.

## Materials and methods

### Study design

This is a non-randomized study assessing the feasibility of the iSupport manual for family caregivers of PWD and formal caregivers in Dhaka, Bangladesh. A quantitative approach will be used to answer the research questions. Quantitative data will be collected at pre- and post-intervention surveys to assess changes in knowledge about dementia care among both family and formal caregivers, as well as changes in psychological well-being among family caregivers. To assess knowledge retention, a follow-up survey will be conducted three months after the post-intervention survey. Moreover, individual qualitative interviews and Focus Group Discussions (FGDs) will be conducted with caregivers after the intervention period to gain understanding of their experiences of using iSupport and possible facilitators and barriers related to using iSupport. The intervention and data collection phases for this study is expected to take around 10 months. This study protocol adheres to the SPIRIT (2013) checklist (S1 File).

### Participant selection and recruitment

**Recruitment and consent process.** Two different recruitment strategies will be parallelly employed to recruit participants for this study. The study aims to recruit family caregivers through contacts with various clinical and professional institutes specializing in neuroscience and mental health. The research team plans to engage with a variety of institutions catering to PWD in Bangladesh, specifically in Dhaka.

Since there is no identified pool from which to recruit family caregivers of PWD, family caregivers will be recruited over a seven-month period (see Fig 1). To facilitate recruitment, the research team will request clinical and professional institutes to disseminate information about the study to potential participants within their networks. Interested individuals will contact the research team and be provided with detailed information about the study objectives and procedures.

Participation will be voluntary, and only those who provide consent will be enrolled in the study. Potential participants will be clearly informed of the objective of the study and the possible risks. Written informed consent will be taken from them after ample time has been given to raise queries and contemplate participation. If necessary, verbal consent will be taken in the presence of a witness, following international guidelines. Additionally, the participants will be made aware of their right to refuse to answer any question and withdraw from the study at any point in time without any consequences.

Formal caregivers will be recruited from an existing pool of caregivers employed at a social enterprise offering formal home care services. This social enterprise is ISO-certified, indicating alignment with international standards in training, labour law compliance and caregiver employment criteria. The recruitment period for formal caregivers is expected to take comparatively shorter time (see Fig 2). The same consent process will be followed during recruitment of formal caregivers. Additionally, they will be made aware that their participation or withdrawal from the study will not affect their employment in any way.

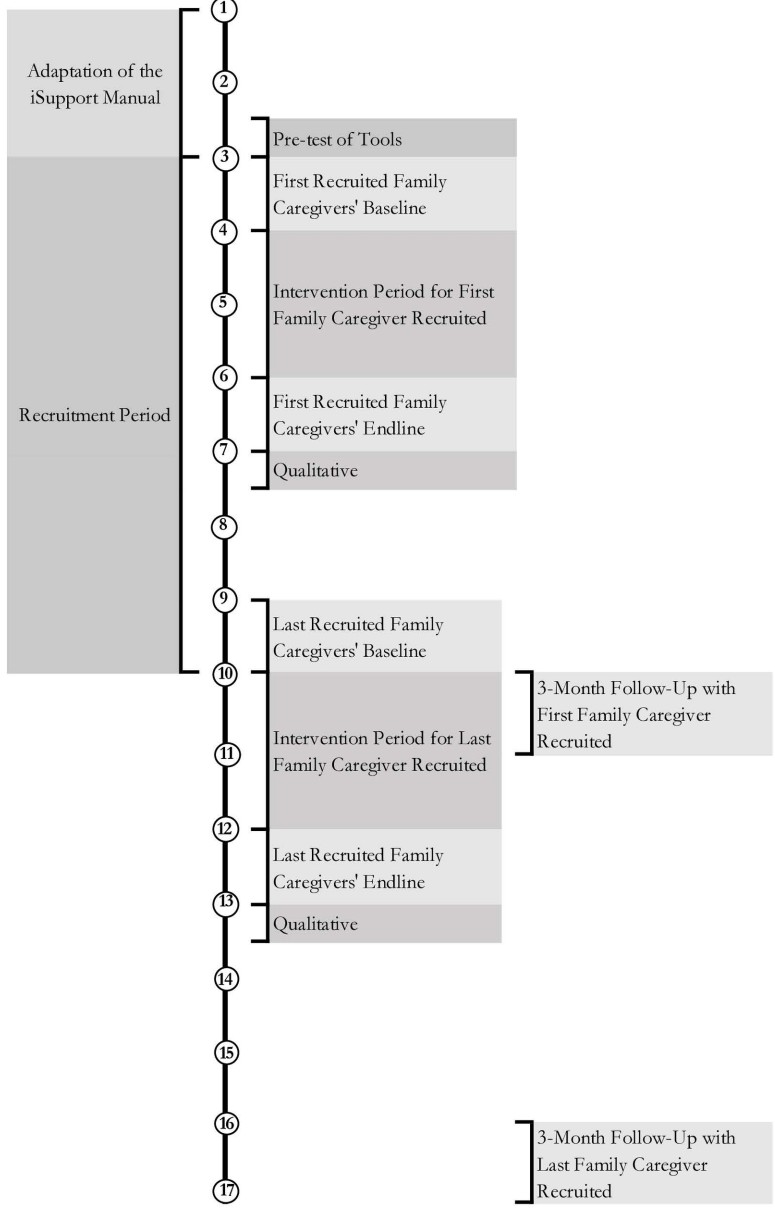

**Fig 1. Timeline (in months) of the project for family caregivers.**

**Eligibility.** Participants eligible for this feasibility study need to be based in Dhaka, Bangladesh and need to fulfil the pre-determined eligibility criteria for either one of the two participant groups, family caregivers or formal caregivers.

To be considered a family caregiver, the participant must be (i) the primary caregiver of PWD, (ii) provide care to a PWD for at least three months, (iii) an unpaid caregiver, (iv) aged 18 years or above, (v) able to read, write and understand Bangla, (vi) able to browse the internet on computers and/or handheld devices, and (vii) able to access the internet at their own cost.

Unlike the family caregivers, formal caregivers are not required to be caring for PWD specifically. However, formal caregivers must be individuals employed by the selected social enterprise and, therefore, meet the enterprise's eligibility criteria

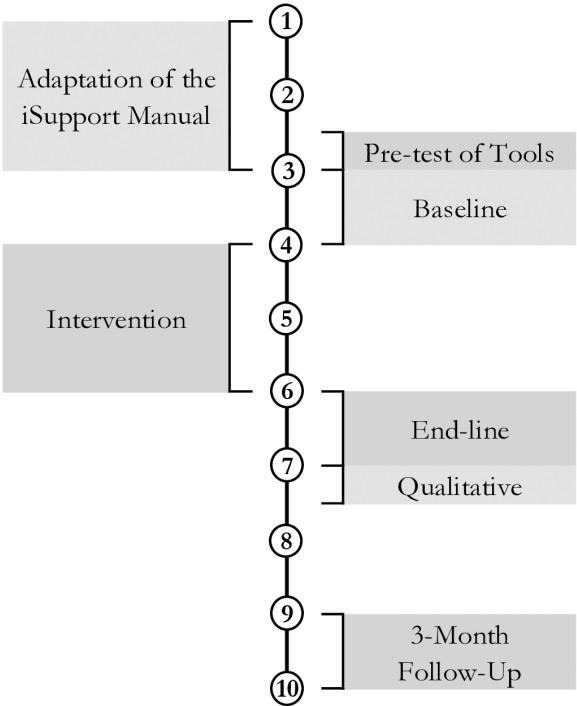

**Fig 2. Timeline (in months) of the project for formal caregivers.**

for formal caregiving roles. These include a minimum of 10 years of schooling for initial employment, and at least one year of experience within the organization for further advancement as a caregiver. Moreover, it is not compulsory for formal caregivers to have access to the internet for this study as the intervention (i.e., iSupport) will be provided both online and in printed-format. Instead, formal caregivers must be (i) aged 18 years or above and (ii) able to read, write and understand Bangla. If any formal caregiver opts for the online manual, then the caregiver must also be (iii) able to browse the internet on computers and/or handheld devices, and (iv) able to access the internet at their own cost, in addition to the previous eligibility criteria. Formal caregivers' ability to browse the internet will be assessed in two ways. First, during the baseline survey, they will be asked directly whether they can browse the internet by themselves. Second, their ability to access and navigate the web-based manual will be observed during a live demonstration of the website. During the training session, caregivers will be required to log in to the website themselves, while the trainer will observe their ability to navigate its features. Formal caregivers with limited or no access to the internet will receive a printed format of the iSupport manual.

## The iSupport intervention

The iSupport manual consists of five modules, each module sub-divided into lessons, accompanied with exercises related to the lessons covered. The content is designed to equip caregivers with comprehensive knowledge and practical skills for supporting PWD and managing associated challenges. The topics covered in the modules include understanding dementia and its impact, practical caregiving techniques, assistance with daily activities, coping strategies for behavioural changes, and stress reduction strategies for caregivers. Both family and formal caregivers will receive the same content. An overview of the modules and lessons of the iSupport manual is provided in Fig 3.

For this research, the Bangladeshi adaptation of the iSupport manual will be uploaded online, where participants must log in using unique log in credentials provided by the research team. The platform will allow participants to access and

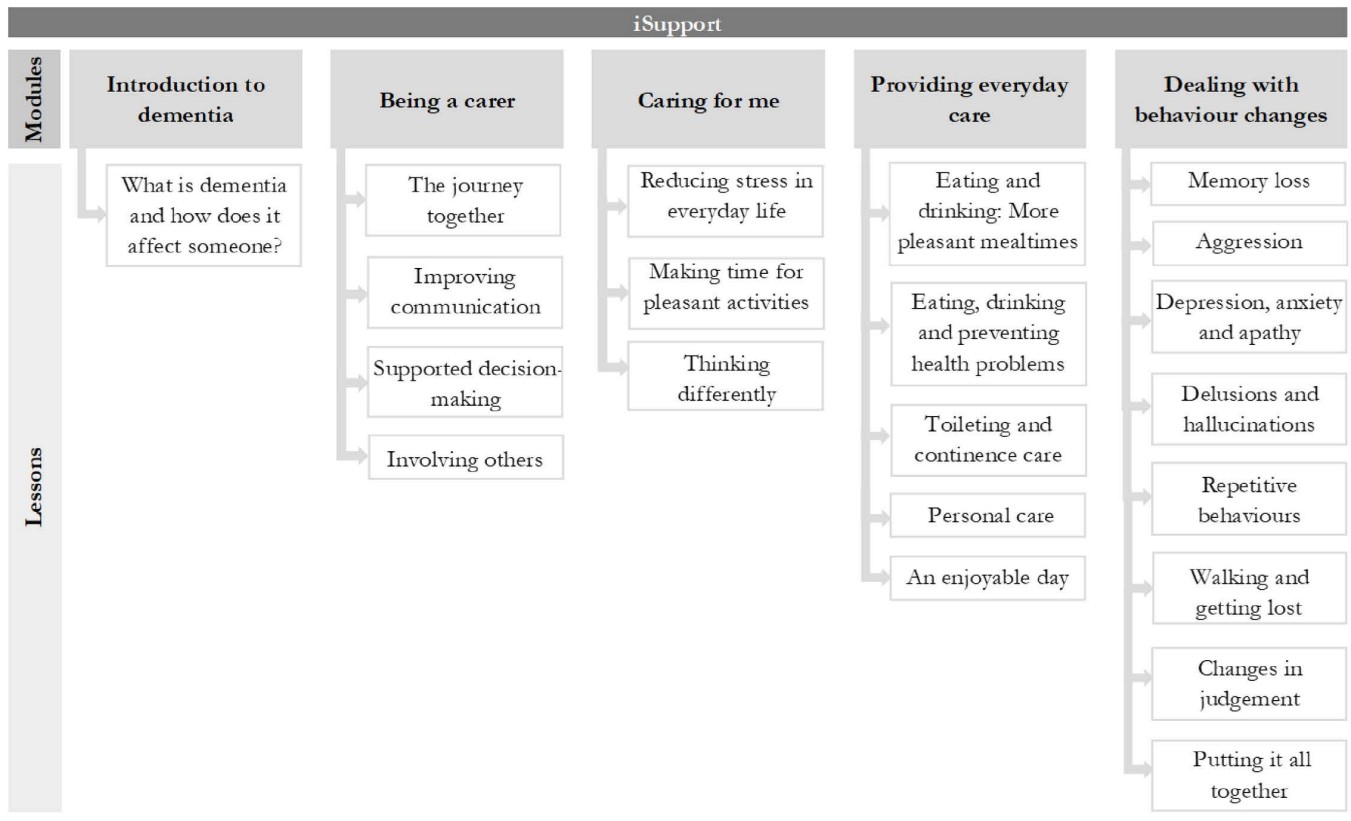

**Fig 3. Overview of the iSupport modules and lessons. Adapted from the WHO's iSupport manual [22].**

navigate modules in any order, add and save personal notes on each page, and complete quizzes at the end of each section. Like the original manual, the answers will be available at the end of each section. The research team can track participants' progress and access saved notes through website analytics.

Both family and formal caregivers will be given access to the website for two months, subsequent to taking part in the baseline survey. Although the iSupport manual was initially designed as a web-based self-help tool, the WHO also states it can be printed and used offline, allowing easy access for users with poor internet connection, no smart phones or other mobile devices, and no access to computers [22]. Therefore, formal caregivers recruited in this study will also have access to printed versions of the manual as they are likely to have limited access to internet or they may not have compatible devices or devices with large enough screens to view the manual. Moreover, formal caregivers will receive face-to-face training on how to use the iSupport manual prior to obtaining access to the manual. Study participants are sent a weekly reminder of their access duration and to encourage engagement with the iSupport manual. Access to, both, online and printed versions will be restricted by disabling online access and taking back the printed manual before their end-line survey.

### Adaptation

To ensure efficacy of the iSupport manual in improving knowledge about dementia and caregiving skills required to provide care for PWD, and reducing stress of caregivers of PWD in Bangladesh, the manual has been translated into Bangla and adapted to the Bangladeshi cultural context. Currently, a Bangla version of the iSupport manual does exist. However,

this manual was adapted for Bangla-speaking communities living in the UK [37]. Given the differences in how Bangla as a language is used in Bangladesh and in the UK, and other distinct cultural aspects particular to the context in Bangladesh, adaptation was necessary for its use by the population in the country.

The adaptation of the Bangla iSupport manual followed a structured four-step process: (1) content translation, (2) linguistic and cultural adaptation, (3) stakeholder engagement and approval, and (4) fidelity assessment. The adaptation process followed the standard guidelines provided by WHO [38], and the research team informed WHO about the adaptation for the Bangladeshi context. International adaptation practices [33,34] were also followed to ensure the manual's relevance and suitability for caregivers in Bangladesh. The adaptation began in November 2023 and concluded in August 2024.

The translation was conducted by researchers with expertise in health research and intervention programs, and fluent in both Bangla and English. The draft versions were then refined through multiple reviews to ensure consistency with the original iSupport manual, including case studies, examples, tips, and activities. Throughout the translation process, any uncertainties or disagreements regarding technical terms were documented and deliberated upon in team meetings until a consensus was reached. The adaptations were systematically documented using WHO's adaptation form, with modifications made to the language, examples and illustrations to align with the local societal norms, caregiving practices and lived experiences.

Once the adaptation form and translation were completed, they were reviewed by experts in dementia care and an individual proficient in Bangla with knowledge on dementia. The adapted manual then underwent further validation through a stakeholder meeting involving the research team, formal caregivers, medical practitioners specializing in dementia care, and other relevant organizations or experts in the field. The stakeholder meeting, including five formal caregivers (two men and three women), resulted in linguistic modifications. Although the original iSupport manual states that the WHO is not responsible for the content or accuracy of this translation [22], all adaptations were documented in structured forms, which were submitted to the WHO for a fidelity check. The WHO assessed the extent to which the adaptations aligned with the original manual's core concepts and messages and advised the research team to submit a case study detailing the challenges faced during the adaptation process.

## Study instruments

Quantitative and qualitative data collection tools will be used to assess the primary and secondary objectives of this study. Bangla version of all data collection instruments will be used for this feasibility study.

Since the primary objective of this feasibility study is to assess the knowledge on dementia and dementia care, the Dementia Knowledge Assessment Scale (DKAS) [39] will be used to measure the change in knowledge among both family and formal caregivers. Caregiving skills will be assessed as one of the secondary objectives for family caregivers. For this purpose, a Caregiver Competency (CC) instrument has been developed by the research team upon reviewing relevant literature and existing tools, and the contents of the iSupport manual.

The level of burden, depression and anxiety experienced by family caregivers will be assessed as secondary objectives as well. This will be done using the Modified Caregiver Strain Index (MCSI) [40], Center for Epidemiological Studies of Depression Short Form (CES-D-10) [41] and the General Anxiety Disorder – 7 (GAD-7) [42]. This study will also assess quality of life of family caregivers by using the care-related Quality of Life Instrument (CarerQoL) [43]. Additionally, family caregivers' relationship with PWD will be assessed using the Quality of Carer-Patient Relationship (QCPR) [44]. It is important to note, while the QCPR instrument is designed to be asked to, both, the caregiver and care-recipient, in this study only the caregivers' response will be recorded. The study instruments mentioned will be administered in Bangla. All tools except GAD-7 and CESD-10 were translated by the research team and pre-tested with caregivers. The Bangla GAD-7 [45,46] and CESD-10 [47] were previously validated in studies conducted in Bangladesh.

Semi-structured interview guidelines will be used to conduct qualitative interviews with the family and formal caregivers. In these qualitative interviews, participants' experiences with iSupport, the benefits they derived from it, the challenges they faced, and potential suggestions to address those challenges will be discussed.

**Sample size**

The primary objective of this study is to assess the effectiveness of the adapted iSupport manual in enhancing the overall knowledge about dementia and dementia care among both family and formal caregivers. The required sample size estimation was, thus, based on the DKAS. The DKAS has previously been used in multiple studies to assess knowledge and awareness of dementia especially among caregivers of PWD [39,48–50].

A hypothesis test on the mean difference between the pre- and post-intervention DKAS scores will be performed. The hypotheses of interest are as follows:

$$H_0 : \mu_d = 0$$

$$H_1 : \mu_d \neq 0$$

where, $\mu_d$ is the mean difference of DKAS scores in the population. The sample size was estimated using the following formula:

$$n = \left( \frac{Z_{1-\alpha/2} + Z_{1-\beta}}{\mu_d/\sigma_d} \right)^2$$

here, σd is the standard deviation of mean differences of DKAS scores in the population, $Z_{1-\alpha/2}$ is the value from the standard normal distribution at the significance level $\alpha$, and $Z_{1-\beta}$ is the value at the power $\beta$.

According to a 2016 study by Annear and colleagues (2017) which validated the DKAS among an international cohort of participants, the mean DKAS score was about 34.5 (in a scale of 50; SD$_{Family}$=6.8 and SD$_{Formal}$=8.4) for both family and formal caregivers [39]. To estimate the sample size, relevant literature that used the DKAS to evaluate knowledge-based interventions related to dementia were consulted. Eccleston and colleagues (2019) evaluated the impact of participating in a Massive Open Online Course (MOOC) on dementia in improving knowledge related to dementia [48]. The course was offered world-wide, however, the majority of the participants (i.e., 74.3%) were from Australia. Their study used DKAS as an outcome indicator as in the proposed study. The published dataset was used to estimate the baseline and end-line DKAS scores [48]. According to the study, the mean difference between overall pre- and post-intervention knowledge scores was about 9.9 (SD=8.7) and the mean score difference among the participants with exposure to dementia from their family was about 15.0 (SD=9.0) [48].

Using these estimates, the sample size was calculated according to the hypotheses of interest at 5% level of significance and 80% power of the test. Since majority of the data represent Australian population, the standard deviation of the mean differences was inflated while estimating the sample sizes. The standard deviation of the mean differences ($\sigma_d$) for the family caregivers was multiplied by a factor of three and $\sigma_d$ for formal caregivers was multiplied by a factor of two. This conservative approach is expected to help the knowledge assessment adjust for higher variation in DKAS scores. Moreover, the $\sigma_d$ for family caregivers was inflated by a higher factor because the family caregivers' DKAS scores are expected to vary more than that of the formal caregivers. Using the formula, the estimated sample size was 26 for both family and formal caregivers. A 40% attrition rate was assumed since dropouts are comparatively high in these interventions [51]. Incorporating for the attrition, the adjusted sample size for this study would be 44.

However, to test whether the impact of the iSupport manual on knowledge about dementia varies based on gender and caregiving skills a larger sample will be required. To test the knowledge related outcome between male and female formal caregivers, a sample size of 88 (i.e., 44 from each of the gender categories) will be required. The formal caregivers are categorised into three levels by their employer: Level 1, Level 2 and Level 3 (i.e., basic-, general-, and advance-care respectively). This classification is based on their practical skills, work ethics, and educational qualifications, with a minimum requirement of a Secondary School Certificate (10 years of schooling). The categorization of formal caregivers are indicators of their skills. Hence, to test whether the outcome varies among these three levels, a sample size of 132 (44 from each level) will be required. Moreover, the large sample size will allow potential differences in dementia knowledge between male and female formal caregivers before and after the intervention to be identified, as well as to explore whether baseline knowledge differs between the two groups. Therefore, a purposive sampling technique will be applied, and all the available formal caregivers from the specific agency offering formal caregiving services will be invited to participate in the survey. Family caregivers will also be purposively selected since there is no identified pool of family caregivers to create a sampling frame.

## Data collection plan

Once the iSupport manual has been fully translated, reviewed, and face-validated through stakeholder workshops, participant recruitment and data collection will commence. Trained data collectors will conduct the data collection. The data collectors will undergo vigorous training on the study instruments prior to data collection. Data collection will begin with a baseline survey, after which participants will gain access to the manual for two months. At the end of the two months, an end-line survey will be conducted. Following the end-line survey, qualitative interviews to explore the experiences, facilitators, and barriers of using iSupport will be conducted. Finally, a follow-up survey will be conducted to test the retention of knowledge on dementia after three months of the end-line survey. Figs 1 and 2 show the adaptation, participant recruitment and data collection timeline of family and formal caregivers.

The DKAS [39] will be used at baseline and end-line to assess the primary objective for both family and formal caregivers. The DKAS will also be used three months after the end-line to assess the retention of knowledge among both caregiver groups. The CC, MCSI [40], CES-D-10 [41], GAD-7 [42], CarerQoL [43] and QCPR [44] will be administered at baseline and end-line only to assess the secondary objectives relating to family caregivers in this study. All data collection tools used will be used to record caregivers' responses only and will not involve the PWD. The timeline for administering each tool is shown in Fig 4.

In addition, baseline data such as gender, age, participants' socio-demographic, economic features, and experience of caregiving will be collected. Website analytics will be collected to assess participants' adherence to the program. Combined with website analytics, a self-reported adherence module at end-line will be used to assess adherence. Moreover, a minimum of 15 individual qualitative interviews with family caregivers and four FGDs with formal caregivers will be conducted after the end-line. Additional interviews and FGDs may need to be conducted until data saturation is reached. Participants with varied degrees of adherence will be invited to participate in the interviews and FGDs.

## Data management

A meticulous data management and monitoring plan will be followed. Data will be collected primarily on SurveyCTO, a secure mobile data collection platform. The datasets generated will be uploaded and securely stored on a password-protected secure server to safeguard unauthorized access. Audio-recordings from qualitative interviews will be deleted from the recording device after they have been copied to a password-protected secure storage device. Moreover, website analytics along with participants' emails used to sign up on the website will be stored.

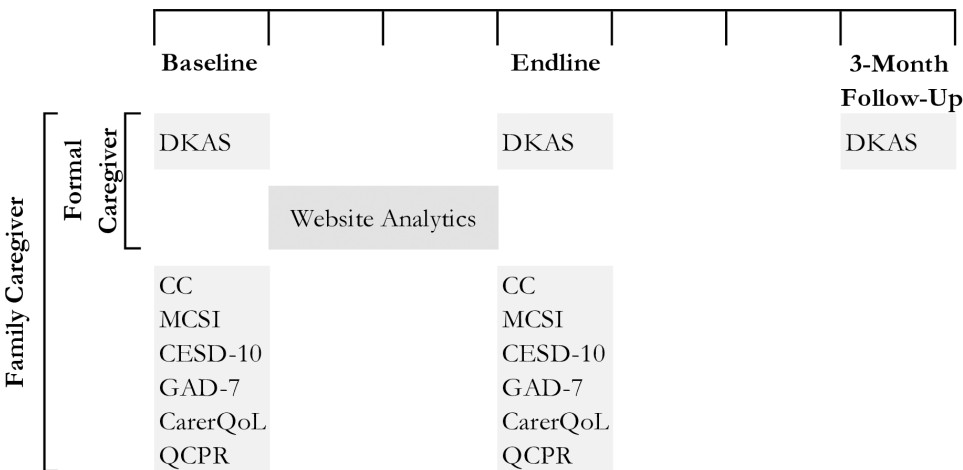

**Fig 4. Data collection and tool administration timeline.**

Hard copies of the consent forms by the study participants will be safely stored in locked and fireproof filing cabinets. Digitized versions of the consent forms will also be stored on a password protected cloud. Study participants will each be pseudonymized with a unique identification number in the database. This unique identification number will be used to facilitate data linkage throughout the study. Only authorized members of the research team will have access to the datasets. De -identified participant data will be made publicly available as supplementary information, accompanying the publication of results, in accordance with ethical guidelines and data protection regulations.

To ensure data accuracy, a rigorous quality assurance process will be implemented. Initial data entry will be cross-checked by a member of the research team, with periodic random data entry checks after. The research team will also conduct weekly meetings to monitor data collection activities and ensure adherence to protocols.

## Data analysis plan

The quantitative results of pre- and post-intervention assessments will be compared. It is hypothesised that statistically significant differences between the pre- and post- outcome indicators will be found. The null hypothesis of zero mean difference will be tested using a two-sample paired t-test for all the outcome indicators. Moreover, multivariable statistical techniques will also be used to control for possible confounders (e.g., gender, age, caregiving skills). Moreover, descriptive and regression analyses will be conducted to identify associations between background and outcome variables. In case of missing data, simple imputation methods will be used. For continuous variables, missing values will be replaced with the mean, while for categorical variables, the mode will be used.

To analyse the qualitative data, content analysis with an inductive approach will be used. The qualitative interviews and FGD recordings will first be transcribed verbatim, ensuring accuracy and completeness. The transcripts will then be translated to English as all interviews will be conducted in the participants' native language, Bangla. The transcripts will be thoroughly read to identify meaning units, codes, and categories [52]. The process of inductive content analysis includes open coding, creating subcategories, and abstraction to categories. Findings will be discussed back and forth among the authors until they reach a consensus, to increase validity. Transferability will be ensured through a detailed description of the data collection process and a sensitivity to the context in which data will be collected. No software will be used for data analysis.

## Study status

The study protocol was reviewed and approved by the Institutional Review Board of SAJIDA Foundation on 8th February 2024 (reference number 2024–005-SFIRB). The research team had conducted one stakeholder meeting with formal caregivers and intends to conduct at least another one with family caregivers of PWD to discuss the adapted iSupport manual and pre-test the tools for the feasibility study. Participant recruitment and data collection have not begun yet and is planned to begin in the last week of July 2024. The recruitment process of family caregivers is currently being discussed with relevant organizations and healthcare professionals.

## Discussion

For this study, the WHO's iSupport manual has been adapted to the Bangladeshi context. This study plans to assess the feasibility and effectiveness of the Bangla iSupport manual in improving dementia knowledge and caregiving skills among family and formal caregivers in Dhaka, Bangladesh. This adaptation is the first Bangladeshi adaptation of the WHO iSupport manual and is anticipated to be among the few caregiving support resources in Bangladesh, that is backed by well-established therapeutic techniques.

Currently, there is a lack of policy on dementia care and guidelines on practice in Bangladesh. Findings of this pioneering study will help structure competence-building for both family and formal caregivers in their role for caring for PWD. This is one of the few studies, if not the first one in Bangladesh, which will help create a knowledge base on care strategies for a PWD living at home and self-care strategies for caregivers themselves.

This research aims to contribute significantly to the limited literature on dementia care in Bangladesh and limited formal training and support services for caregivers. We anticipate that participation in this study will enhance knowledge on dementia among participating caregivers. This in turn may help caregivers better manage PWD [18,19]. This study, however, will not assess the impact of caregivers utilizing the iSupport manual on PWD. Moreover, we anticipate reduced stress and improved mental health outcomes in family caregivers to PWD. This is particularly important in Bangladesh, where mental health support is limited [53].

Through this research, a web-platform and caregiving content will be developed for Bangladeshi Bangla-language speakers in Bangladesh. The research team aims to prepare manuscripts detailing the adaptation process of the iSupport manual, and findings and implications of the intervention. The manuscripts will be submitted to international journals specializing in dementia care, healthcare and caregiver support. The research team also plans to present findings in relevant conferences and seminars. Additionally, the findings of this study and evidence-based recommendations will be communicated with healthcare providers and other organizations working with dementia care through stakeholder meetings and policy briefings.

This feasibility study will provide insights into the feasibility of implementing the iSupport manual in Bangladesh, where caregiving is largely informal, with most PWD receiving care from a family member with little knowledge or paid individual with limited educational qualifications. The findings could inform future adaptations of iSupport for formal caregivers. Lessons learned from the adaptation process, combined with the feedback from stakeholders and participants, will help refine the content and delivery modalities (i.e., online, paper-based, or in-person training) to increase cultural relevance, ease of understanding, and accessibility in resource-constrained settings. These insights will help lay the groundwork for broader trials and scale-up efforts in Bangladesh and other LMICs.

## Limitations

The study will employ a non-randomized design and purposive sampling to recruit participants. Furthermore, all formal caregivers included in the study will be exclusively selected from a single agency. These methodological choices may introduce selection bias and limit the generalizability of findings. The absence of a control group, due to financial and

logistical constraints, will limit the ability to evaluate the effectiveness of the iSupport intervention in achieving the secondary objectives among family caregivers. Moreover, given the iSupport intervention will be online for most participants, internet connectivity and digital literacy may pose as a barrier to proper access and adherence to the intervention. If this occurs, we will be unable to measure the potential effectiveness of the iSupport manual. Use of the printed manual also presents potential challenges, such as the risk of participants photocopying or retaining the manual, which could compromise the accurate assessment of knowledge retention during the follow-up survey.

## Supporting information

**S1 File. SPIRIT checklist.**
(PDF)

## Acknowledgments

The authors are grateful to SAJIDA Foundation for their support in conducting this research. In addition, we thank the caregivers who attended the stakeholder meeting to provide their feedback on the Bangladeshi adaptation of the iSupport manual.

## Author contributions

**Conceptualization:** Muhammed Nazmul Islam, Hanne Konradsen, Marie Tyrrell, Zarina Nahar Kabir.

**Methodology:** Muhammed Nazmul Islam, Hanne Konradsen, Marie Tyrrell, Zarina Nahar Kabir.

**Project administration:** Antara Roy.

**Supervision:** Muhammed Nazmul Islam.

**Writing – original draft:** Muhammed Nazmul Islam, Parsa Musarrat.

**Writing – review & editing:** Muhammed Nazmul Islam, Parsa Musarrat, Hanne Konradsen, Marie Tyrrell, Antara Roy, Asibul Islam Anik, Zarina Nahar Kabir.

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
