## [Decision Letter · Decision Letter 0]

11 Feb 2025

Dear Dr. Islam,

Thank you for submitting your manuscript to PLOS ONE. After careful consideration, we feel that it has merit but does not fully meet PLOS ONE’s publication criteria as it currently stands. Therefore, we invite you to submit a revised version of the manuscript that addresses the points raised during the review process.

We look forward to receiving your revised manuscript.

Kind regards,

Fernanda L. F. Dal Pizzol

Academic Editor

PLOS ONE

Reviewer's Responses to Questions

**Comments to the Author**

1. Does the manuscript provide a valid rationale for the proposed study, with clearly identified and justified research questions?

Reviewer #1: Yes

Reviewer #2: Partly

2. Is the protocol technically sound and planned in a manner that will lead to a meaningful outcome and allow testing the stated hypotheses?

Reviewer #1: Partly

Reviewer #2: Partly

3. Is the methodology feasible and described in sufficient detail to allow the work to be replicable?

Reviewer #1: Yes

Reviewer #2: No

4. Have the authors described where all data underlying the findings will be made available when the study is complete?

Reviewer #1: No

Reviewer #2: No

5. Is the manuscript presented in an intelligible fashion and written in standard English?

Reviewer #1: Yes

Reviewer #2: Yes

You may also provide optional suggestions and comments to authors that they might find helpful in planning their study.

Reviewer #1: The manuscript effectively highlights the growing burden of dementia in LMICs like Bangladesh and the lack of adequate caregiving support. The introduction clearly establishes the need for scalable interventions such as the WHO iSupport manual. The research questions are well-defined, focusing on improving dementia knowledge and caregiving outcomes, and they are justified within the context of the limited literature on dementia care in Bangladesh.

However, while iSupport was originally designed for informal caregivers, the inclusion of formal caregivers in the study is not sufficiently justified, particularly since the content remains unchanged. Formal caregivers might find the examples used in iSupport irrelevant, potentially leading to high attrition or non-adherence, especially given that the eligibility criteria do not require them to be dementia caregivers. Adaptations specific to formal caregivers may be necessary, and as the project is already in progress, the authors should discuss how they plan to address this issue.

Study Design and Methodology

The study protocol outlines a robust methodology that combines quantitative and qualitative approaches. The use of validated tools such as DKAS, MCSI, and GAD-7 enhances credibility. However, several methodological concerns should be addressed:

• The non-randomized sampling method and reliance on purposive sampling may introduce selection bias, potentially limiting generalizability. While the authors acknowledge these as limitations, their impact on the validity of the findings should be discussed more explicitly.

• The recruitment of formal caregivers from a single agency may not capture the diversity of caregiving experiences in Bangladesh. Expanding recruitment to multiple agencies or organizations would improve the representativeness of the sample.

• The eligibility criterion for formal caregivers—“experienced in providing care to older persons”—is vague. Defining a minimum number of years of caregiving experience would improve clarity and ensure consistency in participant selection.

• One of the research questions asks whether the impact of iSupport varies with the level of caregiving skills among formal caregivers. However, the manuscript does not specify how caregiving skill levels will be assessed. A clear operational definition or assessment framework should be provided.

Adaptation of iSupport to Bangladesh

The authors describe the translation and adaptation of iSupport for the Bangladeshi context, including stakeholder consultations and pre-testing. However, the WHO has developed an Adaptation and Implementation Guide (World Health Organization, 2019), which provides structured recommendations for localization. The authors could consider aligning their adaptation process with this guide to ensure cultural and contextual relevance.

Intervention and Digital Accessibility

More details are needed regarding how iSupport will be presented online. For example:

• Will it be structured as an e-book, a blog, or an interactive learning platform?

• How will users navigate different modules and lessons?

• Are there any interactive elements, quizzes, or progress-tracking features?

Additionally, digital literacy and internet access barriers could impact adherence. A brief discussion on how these challenges will be addressed—such as providing offline materials or digital literacy training—would strengthen the feasibility argument.

Statistical Analysis and Data Management

The statistical analysis plan is generally sound, but further details are needed on:

• How missing data will be handled in the analysis.

• The justification for the high attrition rate assumption—providing relevant citations for similar studies would strengthen this estimate.

The manuscript also describes strong data security measures, including the use of SurveyCTO and password-protected storage. However, it does not explicitly state whether the data will be made publicly available post-study. Clarifying the data sharing plan would improve research transparency.

Qualitative Analysis

While the qualitative methods are well-detailed, the manuscript does not specify strategies for ensuring reliability in coding and analysis. Providing details on how themes will be identified, how inter-coder reliability will be maintained, and whether software will be used for analysis would enhance methodological rigor.

Discussion and Broader Implications

The discussion section primarily focuses on justifying the methodology and acknowledging potential challenges. However, it could be strengthened by:

• Expanding on the broader implications of the findings for dementia care policy and practice in Bangladesh.

• Addressing how the study results could inform future adaptations of iSupport or similar interventions in other LMICs.

Ethical and Financial Disclosure

The financial disclosure statement appears contradictory to the acknowledgment section. This should be clarified to ensure consistency.

Overall, this is a well-constructed study protocol that addresses a significant gap in dementia care research in Bangladesh. The study's potential impact is notable, but addressing the outlined concerns—particularly regarding formal caregiver inclusion, intervention delivery, data transparency, and methodological details—will further enhance its rigor and relevance.

Reviewer #2: This study aims to assess the feasibility of using the WHO iSupport (printed) manual with family and formal carers in Bangladesh. There is no mention how the manual would be used in an "online" program as mentioned in the paper. It is also not clear how the manual would be translated into the local language and culturally adapted. Has permission from the WHO been sought for this adaptation? Possibly the cultural adaptation needs to be tested in a pilot study in a smaller sample of carers and professionals, to see if is understandable and useable, using qualitative methods.

The primary aim of the study is to see if iSupport improves the carers knowledge about dementia. While it is clear that the family carers are managing PWD, it is unclear who the formal carers are, and if they are actually managing PWD. It is also not clear why the authors are recruiting a larger sample of formal carers and why they are looking for male vs female differences in the outcome. The two samples could be equal, based on the sample size calculation?

There is no mention about the translation and validation of all the study instruments into the local language. No references are provided if this has already been done before the study.

It is not clear why the iSupport manual would be available to the carers only for two months. What is the reason for this restriction, if at all, and how would the researchers ensure that the manual is not available after this two month period? Without a control group it may not be possible to study the feasibility or "effectiveness" of using the iSupport to test the secondary aims of improvement in caregiver skills, competency, distress, quality of life, etc.

In the consent form there is mention that the formal carers would be monetarily compensated while the family carers would receive no money or incentives. No mention or reason for this is given in the paper itself.

**Do you want your identity to be public for this peer review?** For information about this choice, including consent withdrawal, please see our Privacy Policy

Reviewer #1: **Yes: ** Upasana Baruah

Reviewer #2: **Yes: ** Mathew Varghese

---

## [Author Response · Author response to Decision Letter 1]

6 Apr 2025

Response to Reviewer #1

The manuscript effectively highlights the growing burden of dementia in LMICs like Bangladesh and the lack of adequate caregiving support. The introduction clearly establishes the need for scalable interventions such as the WHO iSupport manual. The research questions are well-defined, focusing on improving dementia knowledge and caregiving outcomes, and they are justified within the context of the limited literature on dementia care in Bangladesh.

Comment 1: However, while iSupport was originally designed for informal caregivers, the inclusion of formal caregivers in the study is not sufficiently justified, particularly since the content remains unchanged. Formal caregivers might find the examples used in iSupport irrelevant, potentially leading to high attrition or non-adherence, especially given that the eligibility criteria do not require them to be dementia caregivers. Adaptations specific to formal caregivers may be necessary, and as the project is already in progress, the authors should discuss how they plan to address this issue.

Response 1: Thank you for your valuable comment. We acknowledge that the content in the iSupport manual may be less relevant for formal caregivers, as the manual was originally designed for family caregivers of persons with dementia (PWD). However, given the limited availability of dementia-specific caregiver training in Bangladesh and of trained caregivers, we believe that iSupport can serve as a valuable introductory resource to improve caregivers' knowledge and understanding of dementia care. Additionally, their employer noted that some formal caregivers already have experience working with PWD. Training can further enhance their ability to recognize dementia symptoms in the people they care for and manage them more effectively. Given the fact that most of their clients are older persons, and the higher prevalence of dementia among older adults in Bangladesh, testing iSupport’s feasibility among formal caregivers can be beneficial.

To maintain engagement and mitigate the risk of non-adherence, a weekly text message is sent on WhatsApp as a gentle reminder to formal caregivers. This is now mentioned in the section describing the iSupport intervention (lines 226-227, page 12 of the revised manuscript with track changes) under Materials and methods.

Comment 2: Study Design and Methodology

The study protocol outlines a robust methodology that combines quantitative and qualitative approaches. The use of validated tools such as DKAS, MCSI, and GAD-7 enhances credibility. However, several methodological concerns should be addressed:

Comment 2.1. The non-randomized sampling method and reliance on purposive sampling may introduce selection bias, potentially limiting generalizability. While the authors acknowledge these as limitations, their impact on the validity of the findings should be discussed more explicitly.

Response 2.1: Thank you for your insightful comment. We acknowledge that the non-randomized design and purposive sampling method may introduce selection bias, which could limit the generalizability of our findings. This has now been mentioned in the Limitations sub-section (lines 482-483, page 24 of the revised manuscript with track changes) in the Discussion section.

Comment 2.2. The recruitment of formal caregivers from a single agency may not capture the diversity of caregiving experiences in Bangladesh. Expanding recruitment to multiple agencies or organizations would improve the representativeness of the sample.

Response 2.2: We agree with the reviewer’s observation. However, expanding recruitment to multiple agencies may not be feasible due to financial and logistical constraints. Moreover, the iSupport intervention involves an extensive reading material and use of an online platform, requiring participants to possess a certain level of education and digital literacy. To our knowledge, there are 26 fully operational caregiving service providers in Bangladesh, with only three providing similar level of service in Bangladesh.

Comment 2.3. The eligibility criterion for formal caregivers—“experienced in providing care to older persons”—is vague. Defining a minimum number of years of caregiving experience would improve clarity and ensure consistency in participant selection.

Response 2.3: Thank you for your valuable comment. We have also reconsidered this criterion and removed it. This change is reflected in the Eligibility sub-section of Participant selection and recruitment (line 190, page 10 of the revised manuscript with track changes) under Materials and methods.

Comment 2.4. One of the research questions asks whether the impact of iSupport varies with the level of caregiving skills among formal caregivers. However, the manuscript does not specify how caregiving skill levels will be assessed. A clear operational definition or assessment framework should be provided.

Response 2.4: The agency from which formal caregivers were recruited classifies their caregivers into three levels: Level 1 (basic-care), Level 2 (general-care), and Level 3 (advanced-care). This classification is based on their practical skills, work ethics, and educational qualifications, with a minimum requirement of a Secondary School Certificate (10 years of schooling). The categorization of formal caregivers are indicators of their skills. This is now indicated in the Sample size section under Materials and methods (lines 359-363, page 18 of the revised manuscript with track changes).

Comment 3: Adaptation of iSupport to Bangladesh

The authors describe the translation and adaptation of iSupport for the Bangladeshi context, including stakeholder consultations and pre-testing. However, the WHO has developed an Adaptation and Implementation Guide (World Health Organization, 2019), which provides structured recommendations for localization. The authors could consider aligning their adaptation process with this guide to ensure cultural and contextual relevance.

Response 3: The adaptation of the Bangla iSupport manual followed most of the standard adaptation guidelines by WHO and international adaptation practices. The process began in November 2023 and concluded in August 2024. The adaptation process involved: (1) content translation, (2) linguistic and cultural adaptation, (3) stakeholder engagement and approval, and (4) fidelity assessment. The cultural adaptation process has now been detailed in the Adaptation sub-section of the iSupport intervention section under Materials and methods (lines 238-277, pages 12-14 of the revised manuscript with track changes).

Comment 4: Intervention and Digital Accessibility

More details are needed regarding how iSupport will be presented online. For example:

• Will it be structured as an e-book, a blog, or an interactive learning platform?

• How will users navigate different modules and lessons?

• Are there any interactive elements, quizzes, or progress-tracking features?

Response 4: The iSupport manual is hosted on a dedicated website, where participants must log in using credentials provided by the research team. The platform allows participants to access and navigate modules in any order, add and save personal notes on each page, and complete quizzes at the end of each section. Although this platform does not provide real-time quiz feedback, like the original manual, the answers are available at the end of each section. The saved notes, along with the participants’ progress can be reviewed by the research team. Details on how iSupport will be presented online has now been added in the second paragraph of the iSupport intervention section in Materials and methods (lines 212-217, page 11 of the revised manuscript with track changes).

Comment 5: Additionally, digital literacy and internet access barriers could impact adherence. A brief discussion on how these challenges will be addressed—such as providing offline materials or digital literacy training—would strengthen the feasibility argument.

Response 5: To address potential digital literacy and internet access barriers, caregivers will be asked during the interview about their internet access, the type of devices they own, and their ability to use the internet independently. Additionally, formal caregivers will receive a brief training session after their baseline survey where a member of the research team will demonstrate how to use the website (mentioned in lines 194-198 on pages 10-11, and lines 224-226 on page 12 of the revised manuscript with track changes). During the training, formal caregivers will be asked to log in to the website themselves and the trainer will observe their ability to navigate the website. If needed, they are given a paper-based version of the manual (mentioned in line 199-200, page 11 of the revised manuscript with track changes). This process of assessing and addressing challenges of digital literacy is detailed in the Eligibility section under Participant selection and recruitment within Materials and methods (lines 194-200 and 224-226 on pages 10-12 of the revised manuscript with track changes).

For family caregivers, digital literacy and internet access are prerequisites for participation. Our eligibility criteria specify that participants must be able to browse the internet on computers and/or handheld devices and have access to the internet at their own cost. These criteria are outlined in under the Eligibility section in Participant selection and recruitment within Materials and methods (lines 185-186, page 10 of the revised manuscript with track changes).

Comment 6: Statistical Analysis and Data Management

The statistical analysis plan is generally sound, but further details are needed on:

Comment 6.1. How missing data will be handled in the analysis.

Response 6.1: Thank you for your comment. Since our survey questionnaire does not include any culturally or personally sensitive questions, we anticipate minimal missing data. Missing values will be handled using simple imputation techniques. For continuous variables, missing values will be replaced with the mean, while for categorical variables, the modal category will be used. This is now mentioned in the Data analysis plan in Materials and methods (lines 424-425 on page 21 of the revised manuscript with track changes).

Comment 6.2. The justification for the high attrition rate assumption—providing relevant citations for similar studies would strengthen this estimate.

Response 6.2: Thank you for your comment. We have cited a study protocol to test the impact of the iSupport manual in Australia who also assumed a 40% attrition rate. The citation has been added in the Sample size section under Materials and methods (line 354, page 18 of the revised manuscript with track changes).

Comment 7: The manuscript also describes strong data security measures, including the use of SurveyCTO and password-protected storage. However, it does not explicitly state whether the data will be made publicly available post-study. Clarifying the data sharing plan would improve research transparency.

Response 7: We appreciate your suggestion regarding data sharing. De-identified participant data will be made publicly available in accordance with ethical guidelines and data protection regulations. The data will be uploaded as supplementary information, with appropriate measures to ensure participant confidentiality. This has now been mentioned in the Data management section under Data collection plan within Materials and methods (lines 410-412, page 20 of the revised manuscript with track changes).

Comment 8: Qualitative Analysis

While the qualitative methods are well-detailed, the manuscript does not specify strategies for ensuring reliability in coding and analysis. Providing details on how themes will be identified, how inter-coder reliability will be maintained, and whether software will be used for analysis would enhance methodological rigor.

Response 8: Thank you for the comment. No software will be used for data analysis. We will analyze data using inductive content analysis. The process of inductive content analysis includes open coding, creating subcategories, and abstraction to categories. Findings will be discussed back and forth among the authors until they reach a consensus, to increase validity. Transferability will be ensured through a detailed description of the data collection process and a sensitivity to the context in which data will be collected. This has now been detailed in the last paragraph of the Data analysis plan under Materials and methods (lines 430-434, page 21 of the revised manuscript with track changes).

Comment 9: Discussion and Broader Implications

The discussion section primarily focuses on justifying the methodology and acknowledging potential challenges. However, it could be strengthened by:

Comment 9.1. Expanding on the broader implications of the findings for dementia care policy and practice in Bangladesh.

Response 9.1: Currently there is a lack of policy on dementia care and guidelines on practice in Bangladesh. Findings of this pioneering study will help structure competence building among family and formal caregivers in their role for caring for PWD. This is one of the few studies or maybe even the first one in Bangladesh which will help to create a knowledge base on how to care for a PWD living at home and even self-care strategies for the caregiver. This text has been added as the second paragraph of the Discussion section (lines 450-454, page 22 of the revised manuscript with track changes).

Comment 9.2. Addressing how the study results could inform future adaptations of iSupport or similar interventions in other LMICs.

Response 9.2: Thank you for your comment. This feasibility study will provide insights into the feasibility of implementing the iSupport manual in Bangladesh, where caregiving is largely informal, with most PWD receiving care from a family member with little knowledge or paid individual with limited educational qualifications. The findings could inform future adaptations of iSupport or similar interventions, including the development of a tailored version of iSupport for formal caregivers. Lessons learned from the adaptation process, combined with feedback from stakeholders and participants, will help refine the content and delivery modalities (i.e., online, paper-based, or in-person training) to increase cultural relevance, ease of understanding and accessibility in resource-constrained settings. These insights will help lay the groundwork for broader trials and scale-up efforts in Bangladesh and other LMICs. This text has been added as the last paragraph of the Discussion section (lines 470-478, page 23 of the revised manuscript with track changes).

Comment 10: Ethical and Financial Disclosure

The financial disclosure statement appears contradictory to the acknowledgment section. This should be clarified to ensure consistency.

Response 10: Thank you for pointing this out. The authors received no external funding for this research project. The research was funded internally by the SAJIDA Foundation. The Acknowledgement section has now been revised to ensure consistency with the financial disclosure statement (see lines 491-492 on page 24 of the revised manuscript with track changes).

Overall, this is a well-constructed study protocol that addresses a significant gap in dementia care research in Bangladesh. The study's potential impact is notable, but addressing the outlined concerns—particularly regarding formal caregiver inclusion, intervention delivery, data transparency, and methodological details—will further enhance its rigor and relevance.

Response to Reviewer #2

Comment 1: This study aims to assess the feasibility of using the WHO iSupport (printed) manual with family and formal carers in Bangladesh. There is no mention how the manual would be used in an "online" program as mentioned in the paper.

Response 1: Thank you for your valuable comment. The iSupport manual is hosted on a dedicated website, where participants must log in using credentials provided by the research team. The platform allows participants to access and navigate modules in any order, add and save personal notes on each page, and complete quizzes at the end

---

## [Decision Letter · Decision Letter 1]

5 May 2025

Dear Dr. Islam,

Thank you for submitting your manuscript to PLOS ONE. After careful consideration, we feel that it has merit but does not fully meet PLOS ONE’s publication criteria as it currently stands. Therefore, we invite you to submit a revised version of the manuscript that addresses the points raised during the review process.

We look forward to receiving your revised manuscript.

Kind regards,

Fernanda L. F. Dal Pizzol

Academic Editor

PLOS ONE

Journal Requirements:

Reviewers' comments:

Reviewer's Responses to Questions

**Comments to the Author**

1. Does the manuscript provide a valid rationale for the proposed study, with clearly identified and justified research questions?

Reviewer #1: Yes

Reviewer #2: Yes

2. Is the protocol technically sound and planned in a manner that will lead to a meaningful outcome and allow testing the stated hypotheses?

Reviewer #1: Yes

Reviewer #2: Yes

3. Is the methodology feasible and described in sufficient detail to allow the work to be replicable?

Reviewer #1: Yes

Reviewer #2: Yes

4. Have the authors described where all data underlying the findings will be made available when the study is complete?

Reviewer #1: No

Reviewer #2: Yes

5. Is the manuscript presented in an intelligible fashion and written in standard English?

Reviewer #1: Yes

Reviewer #2: Yes

You may also provide optional suggestions and comments to authors that they might find helpful in planning their study.

Reviewer #1: The revised version of the manuscript represents a substantial improvement and addresses many of the concerns raised in the initial review. The rationale, methodology, and objectives are clearer, and the adaptation process has been more thoroughly described. However, there are still a few issues that require further clarification and revision to enhance the overall clarity, transparency, and rigor of the study.

Access to Intervention Materials (pg. 12, lines 228-229)

The authors state that “access to, both, online and printed versions will be restricted by disabling online access and taking back the printed manual before their end-line survey.” While the intention is understandable, removing the printed version does not fully guarantee restricted access, as participants could easily photocopy or retain copies of the manual during the intervention period. Unlike the online platform, which is more easily controlled via login access, printed materials cannot be secured in the same way. This should be acknowledged as a limitation of the study in the manuscript.

Clarification Needed-Comment 2.2

In response to comment 2.2, the authors state: “To our knowledge, there are 26 fully operational caregiving service providers in Bangladesh, with only three providing similar level of service in Bangladesh.” The meaning of “similar level of service” is unclear. It would be helpful to specify what is meant by this - does it refer to the quality of training, type of clientele, scope of services, or caregiver qualifications? Please revise this sentence for clarity.

Definition of Formal Caregivers (Comment 2.3)

The authors’ response to the concern about vague eligibility criteria for formal caregivers is not satisfactory. Removing the criterion is not an adequate solution or justification. Instead, the authors should consider incorporating a clearer and stricter definition of “formal caregivers,” such as minimum years of experience, specific care responsibilities, or formal employment status in caregiving services.

Sentence Clarity (pg. 12, lines 238–241)

The following sentence is clunky and difficult to follow:

“This process followed the standard adaptation guidelines provided by WHO (38) upon informing them about the adaptation of the iSupport manual for the specific context of Bangladesh, and international adaptation practices (33,34) to ensure the manual’s suitability for Bangladeshi caregivers.”

I suggest restructuring it for clarity. For example:

“The adaptation process followed the standard guidelines provided by WHO (38), and the research team informed WHO about the adaptation for the Bangladeshi context. International adaptation practices (33,34) were also followed to ensure the manual’s relevance and suitability for caregivers in Bangladesh.”

Data Availability Statement

The authors mention that de-identified participant data will be made publicly available as supplementary information. However, it remains unclear where this data will be published- whether as supplementary files alongside a results publication, in a specific data repository, or on an institutional platform. Please clarify the planned location and format for data sharing to meet transparency and open science standards.

Conclusion

This protocol presents an important study that fills a critical gap in dementia caregiver support research in Bangladesh. If the remaining issues listed above are addressed, particularly those relating to clarity, definitions, and transparency, the manuscript will be significantly strengthened.

Reviewer #2: Reviewer comments have been addressed adequately. No further comments to the authors.

xxxxxxxxxxxxxx

**Do you want your identity to be public for this peer review?** For information about this choice, including consent withdrawal, please see our Privacy Policy

Reviewer #1: **Yes: ** Upasana Baruah

Reviewer #2: **Yes: ** Mathew Varghese

---

## [Author Response · Author response to Decision Letter 2]

19 May 2025

Response to Reviewer #1

Comment 1: Access to Intervention Materials (pg. 12, lines 228-229)

The authors state that “access to, both, online and printed versions will be restricted by disabling online access and taking back the printed manual before their end-line survey.” While the intention is understandable, removing the printed version does not fully guarantee restricted access, as participants could easily photocopy or retain copies of the manual during the intervention period. Unlike the online platform, which is more easily controlled via login access, printed materials cannot be secured in the same way. This should be acknowledged as a limitation of the study in the manuscript.

Response 1: Thank you for your valuable comment. We acknowledge the possibility of participants photocopying or retaining copies of the printed manual. We have now mentioned this as a limitation in the last sentence of the Limitations sub-section (lines 465-467, page 23 of the revised manuscript with track changes).

Comment 2: Clarification Needed-Comment 2.2

In response to comment 2.2, the authors state: “To our knowledge, there are 26 fully operational caregiving service providers in Bangladesh, with only three providing similar level of service in Bangladesh.” The meaning of “similar level of service” is unclear. It would be helpful to specify what is meant by this - does it refer to the quality of training, type of clientele, scope of services, or caregiver qualifications? Please revise this sentence for clarity.

Response 2: “Similar level of service” refers to the quality of caregiver training, overall scope of services provided, and caregiver qualifications. The social enterprise through which the formal caregivers are recruited is the only ISO-certified caregiving agency in Bangladesh, indicating alignment with international standards in training and labour law compliance. Moreover, the enterprise has strict criteria for employment and progress to higher levels, such as minimum 10 years of schooling, minimum one year experience at the agency, Diploma in Nursing, and performance evaluations, which ensure higher standard of caregiver qualifications and scope of service. This has now been briefly mentioned in the Recruitment and consent process sub-section of Participant selection & recruitment (lines 173-175, pages 9-10 of the revised manuscript with track changes) under Materials and methods.

Comment 3: Definition of Formal Caregivers (Comment 2.3)

The authors’ response to the concern about vague eligibility criteria for formal caregivers is not satisfactory. Removing the criterion is not an adequate solution or justification. Instead, the authors should consider incorporating a clearer and stricter definition of “formal caregivers,” such as minimum years of experience, specific care responsibilities, or formal employment status in caregiving services.

Response 3: In this study, formal caregivers are defined by their employment status – specifically, as individuals employed by the selected social enterprise. Their eligibility as formal caregivers are further defined by the enterprise’s criteria, which include minimum 10 years of schooling to become a caregiver, and at least one year of experience within the organization for advanced care responsibilities. This has now been briefly mentioned in the Eligibility sub-section of Participant selection and recruitment (lines 189-192, page 10 of the revised manuscript with track changes) under Materials and methods.

Comment 4: Sentence Clarity (pg. 12, lines 238–241)

The following sentence is clunky and difficult to follow:

“This process followed the standard adaptation guidelines provided by WHO (38) upon informing them about the adaptation of the iSupport manual for the specific context of Bangladesh, and international adaptation practices (33,34) to ensure the manual’s suitability for Bangladeshi caregivers.”

I suggest restructuring it for clarity. For example:

“The adaptation process followed the standard guidelines provided by WHO (38), and the research team informed WHO about the adaptation for the Bangladeshi context. International adaptation practices (33,34) were also followed to ensure the manual’s relevance and suitability for caregivers in Bangladesh.”

Response 4: Thank you for your suggestion. Accordingly, lines 244-248 on page 13 have been restructured in the Adaptation sub-section of The iSupport intervention section of Materials and methods.

Comment 5: Data Availability Statement

The authors mention that de-identified participant data will be made publicly available as supplementary information. However, it remains unclear where this data will be published- whether as supplementary files alongside a results publication, in a specific data repository, or on an institutional platform. Please clarify the planned location and format for data sharing to meet transparency and open science standards.

Response 5: Thank you for your comment. Data will be made publicly available as supplementary files in CSV format, accompanying the publication(s) of results in journals. We have now clarified this in the Data management section of the Data collection plan under Materials and methods (line 388, page 19 of the revised manuscript with track changes).

---

## [Decision Letter · Decision Letter 2]

27 May 2025

Impact of iSupport on improving knowledge on dementia and dementia care among family caregivers to persons with dementia and formal caregivers in Bangladesh: Protocol for a non-randomized feasibility study

PONE-D-24-22636R2

Dear Dr. Islam,

We’re pleased to inform you that your manuscript has been judged scientifically suitable for publication and will be formally accepted for publication once it meets all outstanding technical requirements.

Kind regards,

Fernanda L. F. Dal Pizzol

Academic Editor

PLOS ONE

Additional Editor Comments (optional):

Reviewers' comments:

Reviewer's Responses to Questions

**Comments to the Author**

1. Does the manuscript provide a valid rationale for the proposed study, with clearly identified and justified research questions?

Reviewer #1: Yes

Reviewer #2: Yes

2. Is the protocol technically sound and planned in a manner that will lead to a meaningful outcome and allow testing the stated hypotheses?

Reviewer #1: Yes

Reviewer #2: Yes

3. Is the methodology feasible and described in sufficient detail to allow the work to be replicable?

Reviewer #1: Yes

Reviewer #2: Yes

4. Have the authors described where all data underlying the findings will be made available when the study is complete?

Reviewer #1: Yes

Reviewer #2: Yes

5. Is the manuscript presented in an intelligible fashion and written in standard English?

Reviewer #1: Yes

Reviewer #2: Yes

You may also provide optional suggestions and comments to authors that they might find helpful in planning their study.

Reviewer #1: My previous comments have been satisfactorily addressed and suggestions incorporated. I have no further comments

Reviewer #2: Reviewers comments have been addressed adequately. The article has already been reviewed three times.

**Do you want your identity to be public for this peer review?** For information about this choice, including consent withdrawal, please see our Privacy Policy

Reviewer #1: **Yes: ** Dr. Upasana Baruah

Reviewer #2: **Yes: ** Mathew Varghese

---

## [Editor Report · Acceptance letter]

PONE-D-24-22636R2

PLOS ONE

Dear Dr. Islam,

I'm pleased to inform you that your manuscript has been deemed suitable for publication in PLOS ONE. Congratulations! Your manuscript is now being handed over to our production team.

Kind regards,

on behalf of

Dr. Fernanda L. F. Dal Pizzol

Academic Editor

PLOS ONE